# Auto-Rotating Neural Networks: An Alternative Approach for Preventing Vanishing Gradients

## Abstract

Neural networks with saturating activations are often not used due to vanishing gradients. This problem is frequently tackled using Batch Normalization techniques, but we propose to use a different approach: the Auto-Rotation (AR). An existing AR-based method is the Auto-Rotating Perceptron (ARP), which enhances Rosenblatt's Perceptron and alleviates vanishing gradients by limiting the pre-activation to a region where the neurons do not saturate. However, this method is only defined for dense layers and requires additional hyperparameter tuning. In this paper, we present an extension of the ARP concept: the Auto-Rotating Neural Networks (ARNN). With them, we have convolutional layers and learnable pre-activation saturation regions. Regarding our experiments, in all of them we got that the AR outperforms the Batch Normalization approach in terms of preventing vanishing gradients. Also, our results show that the AR enhances the performance of convolutional nets that use saturated activations, even allowing them to slightly outperform ReLU-activated models. Besides that, by activating the AR we get faster convergence and, due to less hyperparameter tuning, we obtain greater ease of use. Furthermore, with our method we experimentally obtained much more uniform and stable gradients (across the layers and epochs, respectively). We expect that our Auto-Rotating layers will be used for deeper models with saturating and non-saturating activations, since our approach prevents vanishing gradients and issues related to gradient continuity, like what occurs with ReLUs.

## 1 Introduction

Artificial neural networks are state of the art for many tasks. They are built by arranging, in a layered fashion, individual units such as perceptrons (see Figure 1) or convolutional kernels. Nevertheless, their design is based not on intuition but on what works best on predictive datasets, without considering biological plausibility or mathematical rigor.

While it is true that there exist other methods for training supervised neural networks (e.g., neuroevolution (Miikkulainen et al., 2019)), the most common approaches to train them are gradient-based methods, like backpropagation. Hence, the properties of the gradients during training have great importance.

In particular, at many modern neural networks, the activation functions employed to introduce non-linear behavior have discontinuous gradients, like ReLU (Nair & Hinton, 2010) and some of its variations (e.g., PReLU (He et al., 2015) and Leaky ReLU (Xu et al., 2015)). These functions are used because they are non-saturating, which means that they allow gradients to not decrease too fast with layer depth as other activations do. However, they have their own set of issues, like the dead ReLU problem (Lu et al., 2019) or discontinuous gradients that hinder the optimization of the loss function during training (LeCun et al., 2015).

Sigmoid and hyperbolic tangent are two saturating activations that are older than ReLU but are often not used in hidden layers due to vanishing gradients. It is still interesting to employ them, because they generate smoother network outputs than when using ReLU (due to the lack of the sparsification (LeCun et al., 2015; Glorot et al., 2011) obtained when ReLU generates a true zero output). This smoothness property is desirable in some applications like regression and control tasks (Parisini & Zoppoli, 1994).

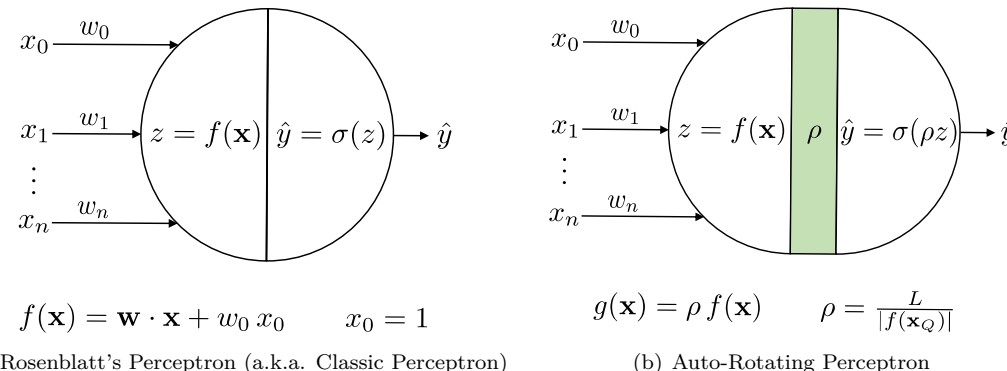

(a) Rosenblatt's Perceptron (a.k.a. Classic Perceptron)  (b) Auto-Rotating Perceptron

Figure 1: Internal structure of a classic perceptron (left) and an Auto-Rotating Perceptron (right). On both neural units, there are two phases: a linear transformation and a non-linear activation. However, on the ARP there is an additional intermediate phase in which the coefficient $\rho$ is computed.

On the other hand, despite all the advancements and research on Deep Learning, notice that the perceptron unit has not been changed since it was formulated by Rosenblatt (1958). Saromo et al. (2019) proposed enhancing the perceptron itself, creating the Auto-Rotating Perceptron (ARP, see Figure 1). This is a novel method designed to alleviate the vanishing gradient problem in saturating activations by constraining the pre-activation to a desired value range in order to minimize saturation of the perceptrons. Previous works with the ARPs show that they can boost neural network learning (Saromo et al., 2020). Nevertheless, this operation is only defined for dense layers to build multilayer perceptrons, and requires tuning of the hyperparameters $\mathbf{x}_Q$ and $L$ (Saromo et al., 2019).

In this paper, we extend the Auto-Rotation (AR) concept from dense to convolutional layers. Thus, we create the Auto-Rotating Neural Networks. With them, we obtain a positive effect on the predictive performance and on the slope of the learning curves (i.e., loss and accuracy). Additionally, we enhance the ARP operation to allow for the Auto-Rotating hyperparameters to be automatically calculated and adjusted.

The contributions of this paper are: 1) To deeply develop the ARP structure and improve its design by automatically calculating the formerly hyperparameters $\mathbf{x}_Q$ and $L$. 2) To present three new $L$ types. 3) To extend the Auto-Rotating operation to convolutional layers (hence, obtaining the Auto-Rotating Neural Networks). 4) Results that show the ARP operation successfully diminishes vanishing gradients. 5) Results that show that by activating the AR we obtain better gradient properties: more uniformity across the network layers, and more stability through the training epochs.

In this paper we claim the following: 1) The ARP reduces vanishing gradients that appear when using saturating activations. 2) Using the Auto-Rotation along with Batch Normalization (BN) results in less vanishing gradients, w.r.t. not using BN or using BN without AR. 3) Using the AR can lead to better performance in terms of obtaining better loss, accuracy, and convergence times; but at a cost of the extra computations related to the $\rho$ coefficient.

This paper is highly relevant to practitioners that build artificial neural network models for tasks that benefit from activations that lead to non-sparse representations. For instance, a saturated activation like the sigmoid is useful for allowing the network to behave like a smooth function, which is useful for regression and feature learning tasks.

## 2   Preliminaries

In this section, we discuss concepts related to artificial neural networks as a background review and nomenclature presentation for then moving to the ideas and contributions of the paper. Additionally, we present our concept of *dynamic region*.

## 2.1 Classic Perceptrons

A perceptron unit (Rosenblatt, 1958) is a function that maps an input vector $\mathbf{x} = \langle x_1, x_2, \cdots, x_n \rangle \in \mathbb{R}^n$ to an output $\hat{y} \in \mathbb{R}$. This mapping is done in two phases, as shown in Figure 1. On the first step, a weighted sum of the inputs is computed and a bias is added: $f(\mathbf{x}) \coloneqq \mathbf{w} \cdot \mathbf{x} + w_0 \, x_0$, where $\mathbf{w} \in \mathbb{R}^n$ is the weight vector, $w_0 \in \mathbb{R}$ is the bias, and the constant $x_0 \in \mathbb{R}$ is set as $x_0 \coloneqq 1$. Also, the bias is sometimes represented as $b$, where $b \coloneqq w_0 \, x_0$. Notice that we can consider the scalar value $z$ as a numerical measurement of the abstract feature extracted by the neuron. On the second step, a non-linear function $\sigma(z)$ (i.e., the activation function) is applied to the weighted sum $z$ to obtain the perceptron output $\hat{y} \coloneqq \sigma(f(\mathbf{x}))$, which after the training is expected to be similar to $y \in \mathbb{R}$.

Because of their layered arrangement inside the network, neurons are hierarchical feature extractors. This functionality is achieved by tuning the learnable parameters $\mathbf{w}$ and $w_0$. Those values are adjusted during the training to make the perceptron output $\hat{y}$ as close to the desired target $y$ as possible. Theoretically, the more number of layers a neural network has (i.e., if we increase the depth of the neural model), the better learning capabilities it obtains (LeCun et al., 2015). However, in practice, training deep neural networks is difficult (He et al., 2016; Bengio et al., 2009).

## 2.2 Vanishing Gradient Problem (VGP)

While it is true that by stacking more layers the neural model can learn more complex patterns from the input data, a problem arises when training deep neural networks using gradient-based methods: the Vanishing Gradient Problem (VGP) (Bengio et al., 1994; Hochreiter et al., 2001). It appears while adjusting models that use saturated activations (e.g., sigmoid), and also at ReLU-activated models (i.e., dead ReLU (Lu et al., 2019) when $z < 0$.

In an artificial neural network, layers closer to the input learn slower than those near the output (Nielsen, 2015). The reason of this problem is the exponential decrease of the error gradients as we get closer to the input layer during back-propagation learning (Nielsen, 2015; Xu et al., 2016).

To better understand the VGP, recall that the error gradient depends on the derivative $\sigma'(z)$ of the activation function $\sigma(z)$. Hence, when we have more layers, we multiply many times a quantity that depends on that derivative value. Thus, if $\sigma'(z)$ is close to zero, then the VGP appears.

Hence, if we could make the perceptrons to operate in a desired zone which we name the *dynamic region*, the learning performance would improve. This design modification is indeed the core working principle analyzed in this paper.

## 2.3 Dynamic Region of an Activation Function

We formally define the *dynamic region* of an activation function $\sigma(z)$ as a non-unique symmetric (w.r.t. 0) $L$-bounded numerical range (i.e., $z \in [-L, +L]$, with $L > 0$) from where we would like the pre-activation values $z$ to belong in order to avoid node saturation.

This node *dynamization* (i.e., desaturation) is achieved in two ways. First, by preventing the derivative of the activation function to take very small values. Recall that, to prevent the VGP, we do not want the derivative $\sigma'(z)$ of the activation function $\sigma(z)$ to take tiny values. Second, by reducing the maximum number of optimizing steps needed for reaching $z = 0$; hence, making it easier for the extracted feature $z$ to change sign during training. Recall that at each step of the optimization process executed during the model training, we change a bit the weights, which slightly changes $z$.

For example, in the unipolar sigmoid activation function shown in Figure 2, we would like it to only receive values from -3 to +3. We can choose this dynamic region because for inputs whose absolute values are higher than 3, the derivative of the activation is too low. Thus, to obtain that desired dynamic region, the $L$ value chosen is 3. On the other hand, in the ReLU activation function drawn in Figure 2, we can choose that depicted dynamic region to make $z$ not to be too far from zero, and hence we can easier escape a potential dead ReLU situation to improve learning performance.

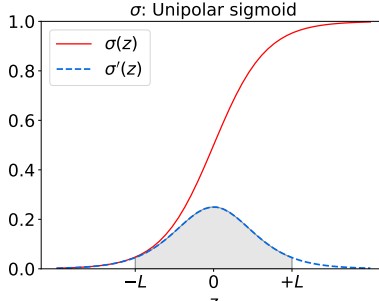 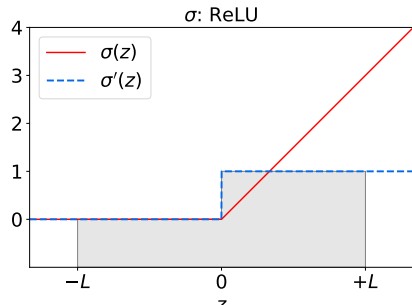

Figure 2: Sigmoid and ReLU activation functions $\sigma(z)$ and their corresponding derivatives $\sigma'(z)$. The projection on $\sigma'(z)$ of a desired dynamic region is depicted in gray.

## 3 From Auto-Rotating Perceptrons to Auto-Rotating Neural Networks

In order to avoid node saturation, the typical solutions are indirect: to change the activation function or to use Batch Normalization (Ioffe & Szegedy, 2015). In this paper, we delve into another approach: to modify the pre-activation phase of the perceptrons so that we can force them to work in a desired dynamic region. Notice that this modification is executed outside the activation chosen for the perceptrons. In other words, we could still change the pre-activation mechanism while using a non-saturated activation function. This direct approach, shown in Figure 1, is called the Auto-Rotating Perceptron (ARP) (Saromo et al., 2019) and was originally proposed only for dense layers. With the intermediate coefficient $\rho$, the ARP operation generalizes and enhances the perceptrons by making the activation function receive values that only come from $-L$ to $+L$, with $L > 0$ (this is the operating principle of the Auto-Rotation).

### 3.1 Delving into the Auto-Rotating Operation

To deeply understand the ARP, we need to define the domain of the input vectors $\mathbf{x} \in \mathbb{R}^n$ that are given to a generic perceptron of a hidden layer. Regarding those input vectors, let $x_i^{(\min)}$ and $x_i^{(\max)}$ be their minimum and maximum scalar components, respectively. These definitions are valid for each $i$-th component of the $n$-dimensional input space. Considering that the input of a perceptron that belongs to a hidden layer is composed of the $\sigma$-activated values provided by the previous layer, and assuming that the data that the input layer receives is restricted to a value range whose limits are known, we obtain that:

$$x_i^{(\min)} := \min\left\{x_i^{(\text{minimum from data})}, \min_{\forall z \in \mathbb{R}} \sigma(z)\right\} \text{ and } x_i^{(\max)} := \max\left\{x_i^{(\text{maximum from data})}, \max_{\forall z \in \mathbb{R}} \sigma(z)\right\}.$$

However, if we use an activation $\sigma(z)$ for which we have one (or both) of these situations: $\min_{\forall z \in \mathbb{R}} \sigma(z) = -\infty$, or $\max_{\forall z \in \mathbb{R}} \sigma(z) = +\infty$; we propose to instead use the following two relations, respectively:

$$x_i^{(\min)} := \min\left\{x_i^{(\text{minimum from data})}, \min_{\forall z \in \mathcal{X}_i} \sigma(z)\right\} \text{ and } x_i^{(\max)} := \max\left\{x_i^{(\text{maximum from data})}, \max_{\forall z \in \mathcal{X}_i} \sigma(z)\right\};$$

where $\mathcal{X}_i$ is defined as $\mathcal{X}_i := \left\{x_i \in \mathbb{R} \,/\, x_i^{(\text{minimum from data})} \le x_i \le x_i^{(\text{maximum from data})}\right\}$.

These equations result in $x_i^{(\min)} \le x_i \le x_i^{(\max)}$, $\forall i \in \{1, 2, \cdots, n\}$; where $x_i^{(\min)}$, $x_i$, and $x_i^{(\max)}$ are the $i$-th scalar elements of $\mathbf{x}_i^{(\min)}$, $\mathbf{x}$, and $\mathbf{x}_i^{(\max)}$ respectively. In other words, $\mathbf{x}^{(\min)}$ and $\mathbf{x}^{(\max)}$ represent the minimum and maximum values that will ever be given to any neuron of a hidden layer. Also, note that these domain limits are not hyperparameters. In fact, they depend on two factors:

- The pre-processing method used for the input dataset. If we apply scaling, then we know beforehand the limits of the pre-processed data. Hence, when using the Auto-Rotation it is strongly suggested to pre-process the dataset using proportional scaling to a defined range, instead of standardizing it.

| Types of $L$ on an Auto-Rotating layer | Shared All neurons in the layer share the same value of $L$ | Indep All neurons in the layer have their (independent) value of $L$ |
|---|---|---|
| **Frozen** $L$ does not change during training | `frozen_shared` | `frozen_indep` |
| **Auto** $L$ is a trainable weight | `auto_shared` | `auto_indep` |

Table 1: Valid types of $L$. There are four possible combinations, depending on: if $L$ is constant or trainable (*frozen / auto*), and if $L$ is shared across the layer or not (*shared / indep*). Notice that `frozen_shared` represents the standard ARP formulation presented in (Saromo et al., 2019).

- The activation function $\sigma(\cdot)$ of the neuron (or layer, if all the neurons in the layer use the same activation).

Besides that, if all the layers of the net share the same activation, then these limits ($\mathbf{x}^{(\min)}$ and $\mathbf{x}^{(\max)}$) can be defined once for the whole network. Also, notice that for regression tasks, the Auto-Rotation must be turned off (only) on the output layer, so that the network outputs can behave as unrestricted real values.

Furthermore, recalling the formulas presented in Figure 1, we can see that if we set $\rho = 1$ on the AR equation, we obtain the formula of the classic perceptron. Hence, the ARPs are a generalization of vanilla perceptrons. Finally, it is important to highlight that the AR is not an activation function, but a pre-activation mechanism applicable to any activation function.

## 3.2 Different Types of $L$

In this paper, we give more flexibility to the AR operation by allowing the user to choose between four different types of the hyperparameter $L$, which are the result of two new different ways of defining it.

First, we start with the original Auto-Rotating formula (Saromo et al., 2019) where $L$ is constant (*frozen* setup), and now we make it automatically tunable by the network during training (*auto* setup).

Second, the original ARP approach indicates having one scalar value of $L$ for the whole layer (*shared* setup). In this paper, we propose changing $L$ so that it can be a set of scalar values independently defined for each one of the perceptrons (*indep* setup). In other words, if the $L$ setup is *indep*, then $L$ becomes a vector whose elements affect the neurons of the Auto-Rotating layer on a one-to-one basis.

By combining these two ways of modifying $L$, we obtain the four $L$ classes shown in Table 1. Also, notice that for all $L$ types, the initial $L$ value (be it a scalar or a vector) must be provided by the user.

Also, when the $L$ setup is *shared*, the $L$ value is the same for the whole layer but the internal hyperplane rotation varies at each neuron (because $\rho$ also depends on $f(\cdot)$). Furthermore, the *auto* setup lowers the impact of the initial $L$ value chosen, because it allows the net to automatically change $L$ to a value that diminishes the prediction error. Thus, we can reduce additional manual tuning of $L$.

On the other hand, we must consider the $L > 0$ restriction. When we make the $L$ value to be learnable (i.e., *auto* setup), the model could assign it a negative value. To comply with the $L > 0$ condition without considerably distorting the original $L$ value, we applied the function $\text{softplus}(x) := ln(1 + e^x) > 0$ to it, in an element-wise fashion. Hence, the effective $L$ used for implementing the AR is $L_{\text{eff}} := \text{softplus}(L)$. Thus, in practice the effective rotation coefficient we use is $\rho_{\text{eff}} := \frac{L_{\text{eff}}}{|f(\mathbf{x}_Q)|}$ instead of $\rho := \frac{L}{|f(\mathbf{x}_Q)|}$. Notice that $\rho_{\text{eff}} \approx \rho$, for $L > 0$ (because for positive $L$ values, $\text{softplus}(L) \approx L$).

### 3.3 The Auto-Rotating layers can now work without extra hyperparameters

Recalling the coefficient $\rho_{\text{eff}} \coloneqq \frac{L_{\text{eff}}}{|f(\mathbf{x}_Q)|} \in \mathbb{R}$ used for the AR, we see that it has two hyperparameters: $\mathbf{x}_Q \in \mathbb{R}^n$ and $L_{\text{eff}} \in \mathbb{R}$. Note that $\mathbf{x}_Q$ has the same shape as the inputs $\mathbf{x} \in \mathbb{R}^n$ given to the neuron.

Regarding the former first hyperparameter, we geometrically obtained that the relation $\mathbf{x}_Q \coloneqq 2\,\mathbf{x}^{(\min)} - \mathbf{x}^{(\max)}$ allows us to automatically calculate it. Thus, $\mathbf{x}_Q$ is not a hyperparameter anymore, and the remaining degree of freedom of the AR layer is controlled by $L$, which defines $L_{\text{eff}}$.

On the other hand, if all the neurons of the layer have the same activation, then $\mathbf{x}_Q$ is made up by identical scalar elements. Hence, in this case we can safely define $\mathbf{x}_Q$ as a scalar and rely on the broadcasting for the calculation of $\rho_{\text{eff}}$. The same idea can be applied for computing $\rho$.

Regarding the former second hyperparameter ($L_{\text{eff}} \coloneqq \text{softplus}(L)$), we tested different scalar values of it under the `frozen_shared` setup. The aim is to find a proper distribution of $L$ values from which we can sample the initial $L$ value (*shared* setup) or values (*indep* setup) to be given to the AR layer as an alternative of manually setting them (more details are provided on Subsection 4.2). This process is analogous to how the initial weights of the layers are sampled from a probability distribution. We propose two probability distributions for $L$ (normal and uniform, see the bottom right part of Figure 3) to be used with any activation and any $L$ type. Thus, when changing from vanilla to AR layers, we now have a configuration that allows the user to choose not to add extra hyperparameters.

### 3.4 Convolutional Auto-Rotation: Conv-ARP

On the formulas $\hat{y} \coloneqq \sigma(\rho_{\text{eff}}\, f(\mathbf{x}))$ and $\rho_{\text{eff}} \coloneqq \frac{L_{\text{eff}}}{|f(\mathbf{x}_Q)|}$ required for implementing the effective Auto-Rotation on dense layers, we see that the function $f(\cdot)$ is a dot product. Hence, we can extend the concept to convolutional layers, because the convolution can be seen as a dot product. Therefore, when we originally talked about the AR inside each one of many perceptrons of a dense layer, now we have an AR inside each one of many kernels that belong to a convolutional layer. Recalling the formulations presented in Subsection 2.1 and observing the equation $f(x) = \mathbf{x} \star \mathbf{w} + b$ (where $\star$ is the convolution operation), we can indeed see a clear similarity with the equations from Figure 1. Thus, an extension of the ARP concept can be done directly by using the same formulation as the standard ARP but with a convolution instead of a dot product.

## 4 Experiments

In this section, we describe the executed experiments to ensure reproducibility. Thereafter, we describe our findings and how they relate to our claims.

### 4.1 Setup

The main objective of our experiments is to show that the Auto-Rotating formulation reduces the VGP, and is competitive with standard batch-normalized models when using saturated activations like the sigmoid. Also, we wanted to analyze the advantages and disadvantages of using different activations and methods to define $L$. We applied the following configuration for all the experiments:

- **Pre-processing:** Scaling of the datasets to the range that goes from 0 to 1, by dividing the inputs over 255 (because all the datasets tested contain only 8-bit images).
- **Input shape:** Fashion MNIST (Xiao et al., 2017) and MNIST (LeCun et al., 1998): ($28 \times 28 \times 1$). CIFAR-10 (Krizhevsky et al., 2009) and SHVN-normal (Netzer et al., 2011): ($32 \times 32 \times 3$). STL-10 (Coates et al., 2011): ($96 \times 96 \times 3$).
- **Neural architecture $\mathcal{A}$:** Conv2D(16, $5 \times 5$, Activation) - BatchNormalization() - MaxPool2D($2 \times 2$) - Conv2D(8, $3 \times 3$, Activation) - BatchNormalization() - MaxPool2D($2 \times 2$) - Flatten() - Dense(15, Activation) - Dense(10, Softmax). No padding at convolutional layers.
- **Neural architecture $\mathcal{B}$:** Conv2D(8, $5 \times 5$, Activation) - BatchNormalization() - MaxPool2D($2 \times 2$) - Conv2D(8, $3 \times 3$, Activation) - BatchNormalization() - MaxPool2D($2 \times 2$) - Conv2D(4, $3 \times 3$, Activation)

- BatchNormalization() - MaxPool2D($2 \times 2$) - Flatten() - Dense(10, Activation) - Dense(10, Activation) - Dense(10, Activation) - Dense(10, Activation) - Dense(10, Softmax). At convolutional layers: zero padding and output has the same size as the input.

At each experiment, we analyzed different types and/or initial values of $L$ but making sure that the initial non-Auto-Rotating weights of the models involved were the same.

We used the Adam optimizer (Kingma & Ba, 2014) (with learning rate $\alpha = 0.001$) to train the models during 50 epochs on the shallower architecture (Arch. $\mathcal{A}$) and 100 epochs on the deeper one (Arch. $\mathcal{B}$), aiming for convergence of the metrics on the test subdataset. In all the experiments, during training we recorded the loss (categorical cross-entropy) and the accuracy (which is the percentage of correctly predicted samples).

Also, when using the ReLU activation for the layers, by using the equations shown in Subsection 3.1 to define $x_i^{(\min)}$ and $x_i^{(\max)}$, we obtained that:

$$x_i^{(\min,\ \sigma:\ \mathrm{ReLU})} := \min\left\{x_i^{(\mathrm{minimum\ from\ data})}, 0\right\} \text{ and } x_i^{(\max,\ \sigma:\ \mathrm{ReLU})} := x_i^{(\mathrm{maximum\ from\ data})}.$$

The library of the Auto-Rotating layers is available in (Anonymous, 2023), along with some use examples. This implementation is based on the Keras library (Chollet et al., 2015).

### 4.2  Effect of the $L$ value

To choose the value of the hyperparameter $L$ (which defines the dynamic region), it is required to analyze the priorly selected activation function and its derivative. In fact, there is a trade-off: with a bigger $L$, you accept more non-linearity for the activation function (which increases the neural plasticity of the model), but at the same time you get more saturation (which slows down the learning).

We tested different scalar values of $L$ (see Figures 3 and 4), to analyze its impact on the neural learning performance. On Figure 3, we can see that for an architecture that uses the unipolar sigmoid activation, the $L$ value that leads to the lowest loss tends to be around $L = 5$. Besides that, we see that the loss and its dispersion worsen when $L > 10$. Hence, we proposed the two $L$ distributions shown in Figure 3. Also, we can see that in all the tested datasets the AR models can outperform the vanilla network. All the experiments executed in this subsection are with the $L$ type set to `frozen_shared`.

### 4.3  Effect of the $L$ types

In the experiments depicted in Figure 4, we first trained the models with the $L$ type set to `frozen_shared`, and gave them the $L$ values $2, 4, 6, 8,$ and None. Then, we selected the non-None $L$ value that led to the lowest loss (which for our experiments is the categorical cross-entropy), as the starting $L$ value used to create models with the same initial weights but employing the remaining three types of $L$: `frozen_indep`, `auto_shared`, and `auto_indep`. The vanilla model (where $L = $ None) is taken as a baseline. Hence, to ease the comparison with the other configurations of $L$, we plot its value as an horizontal dashed line. Also, to have more statistical strength of our claims, we executed each of the experiments five times. In the plots of Figure 4, the middle lines and shades represent the median and interquartile range (IQR), respectively.

Note that by applying the most flexible AR type (`auto_indep`) on the sigmoid-activated nets (see the orange plots on Figure 4), they can outperform their ReLU and Leaky ReLU vanilla counterparts.

### 4.4  Effect on the Gradient Magnitude and interaction with Batch Normalization

The main claim of the Auto-Rotating operation is that it can alleviate the VGP by constraining the elements of the Auto-Rotating layers (perceptrons, at dense layers; and kernels, at convolutional layers) to only operate in a desired dynamic region (which is controlled by the value of $L$). To verify this claim, we recorded the gradients of the non-AR weights. This process was done for each one of the network layers across all training epochs. However, recall that one of most common ways to avoid the VGP is to use BN. Hence, to compare

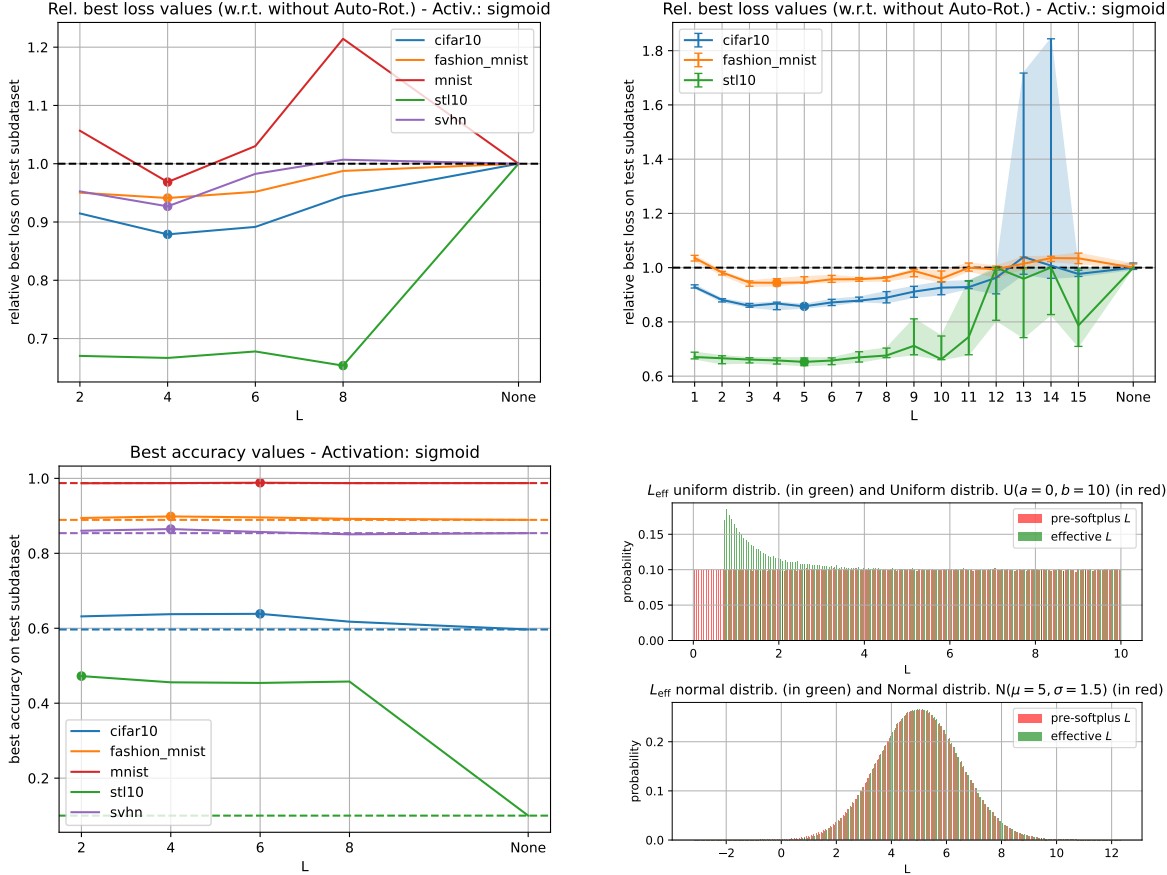

Figure 3: Analysis of the $L$ value. Top left: relative test loss w.r.t. the vanilla model (lower is better), top right: relative test loss on 10 trials (lower is better, median and IQR depicted), and bottom left: test accuracy (higher is better). For the loss and accuracy plots, consider the following: 1) $L$ type: `frozen_shared`. 2) BN: off. 3) Neural Arch.: $\mathcal{A}$. 4) The best metric value found is highlighted with a dot, and the dashed line indicates the metric value related to the vanilla net (where $L = \text{None}$). Bottom right: in red, two distributions proposed for the $L$ value (uniform and normal); and in green, their corresponding $L_{\text{eff}}$ distributions.

BN and AR, we tested the four combinations of activating them or not. We used the deeper neural topology (Arch. $\mathcal{B}$), and ensured having the same initial network weights.

Observe in the experiments done on the CIFAR-10 dataset (see Figure 5) that the vanilla model has indeed much lower gradient values than its AR counterpart. Besides that, we see that using only BN clearly makes the heatmap more red-coloured than the vanilla case, eliminating all the blue and most of the green parts (low gradients). In addition, we observe that using only AR does a much better job on alleviating the VGP than using only BN: with only AR there are absolutely no yellow zones on the later epochs. Interestingly, when using both BN and AR, we obtain the best VGP mitigation: the model does not suffer from the initial stagnation on the upper left part of the heatmap, and there are no yellow areas. Other experiments with similar results are presented in the Appendix.

Furthermore, with AR (with and without BN), we obtain a much more uniform gradient value, which means a more undeviating speed of learning across the layers (observe the variation of the gradients as we move vertically within the heatmaps shown in Figure 5). Besides that, the gradient values are more stable during the training (observe the variation of the gradients by sweeping horizontally to the right, within the heatmaps shown in Figure 5). Notice that with AR, after around four epochs the gradient values converge: it becomes orange very fast. In contrast, without AR, the gradient values present a steep change (at top left heatmap:

from dark blue to orange), and a slower change (at the lower zone of the two left heatmaps: from light-green to a yellow/orange tone). A similar behavior is also seen in all the other experiments (see the Appendix).

Hence, we can graphically see a diminishing of the VGP. This decrease leads to a marginal gain on the learning performance, as depicted in the bottom of Figure 5. However, the advantages gained with the AR are better gradient properties, a faster convergence, and a steeper slope of the loss and accuracy plots.

### 4.5 Discussion and Analysis

We obtained evidence that changing classic (i.e., vanilla) layers to Auto-Rotating layers can lead to an improvement in the learning and predictive performance of neural networks. Also, with our method we diminish the presence of vanishing gradients.

On Figure 5 we can see that the lowest values of the gradient heatmaps corresponding to the non-AR models (without and with BN) are located on the bottom area, which means that the layers closer to the input (related to the bottom region of all heatmaps) learn slower than the layers next to the output (because the learning speed of a layer is related to the module of the gradient (Nielsen, 2015)). By changing the layers from vanilla to Auto-Rotating, but keeping the same initial weights of the networks, we obtain a much more uniform learning speed across all network layers. Because the earlier layers can learn from better characteristics extracted by the layers closer to the output, the predictive performance improves. This outcome can be seen at the loss and accuracy graphs (see the bottom part of Figure 5).

Our experimental results (see Figure 5 and the Appendix) show that AR is better than BN in terms of tackling the VGP, and at allowing all the network layers to learn at a speed that is more uniform (i.e., similar across the layers) and more stable (i.e., low variation through the epochs). Furthermore, the best results were obtained when using the AR along with BN.

Besides that, our experiments varying the $L$ types (see Figure 4) show that learning the $L$ value sometimes outperforms fixing it manually. Also, having isolated $L$ values (`frozen_indep` or `auto_indep` structures) tends to bring an extra accuracy enhancement w.r.t. the standard ARP (i.e., `frozen_shared`) on Fashion MNIST, indicating that the pre-activation saturation region should be set independently for each neuron (or convolutional kernel). Notably, when using AR layers there is also an upgrade in performance of ReLU-activated networks (see the right graph on Figure 4).

## 5 Conclusions and Future Work

In this paper we present the ARNN, which allow adjusting the pre-activation regions to prevent gradient saturation. Additionally, we propose methods to automatically tune that dynamic region by setting $L$ as a trainable weight, and to make this value independent for each element of the layer.

On our experiments, we found that the performance boost due to the AR w.r.t. vanilla models is present in all the datasets tested. Also, observe on the bottom left plot of Figure 3 that we obtained more than a 4x improvement in accuracy using the simplest (i.e., the least flexible) of the $L$ types available (`frozen_shared`).

However, in other experiments, the learning performance upgrade given by the AR is marginal. It is important to notice that the boost is stronger if we have a deeper model and a harder prediction task (which is consistent with the results of (Saromo et al., 2019)). On the other hand, the strength of our method relies on providing a much better diminishing of the VGP when using AR along with BN, instead of only BN. Also, we obtain gradients with better uniformity and stability properties. More tests are needed with more complex networks and datasets to better analyze the AR capabilities.

We hope this work encourages deeper pursuing the research on the AR operation. For instance, further investigation is required in order to extend the AR concept to more layer types (like GRU, LSTM, and others), and to apply the AR to pre-trained neural models. Likewise, an experimental analysis is required for the ARNN on regression tasks. Also, more research is needed for exploring the effects of designing specific $L$ distributions for each activation function.

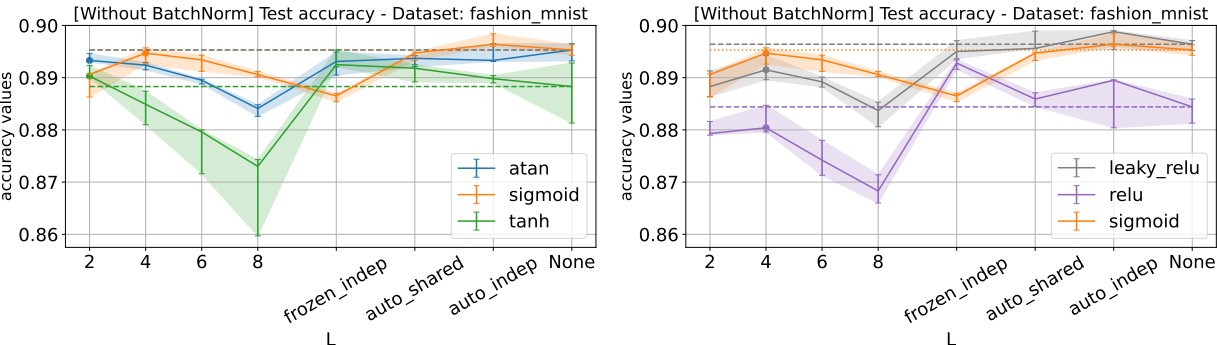

Figure 4: Effect of changing the $L$ type (five trials, Arch: $\mathcal{A}$). First, we executed the training using four `frozen_shared` $L$ values, along with the vanilla model. After that, we highlighted with a dot the non-None $L$ value with the resulting lowest median test loss. Then, we took it as the starting $L$ value used for the remaining three $L$ types. In total, five different sets of initial non-AR weights were used. BN: off.

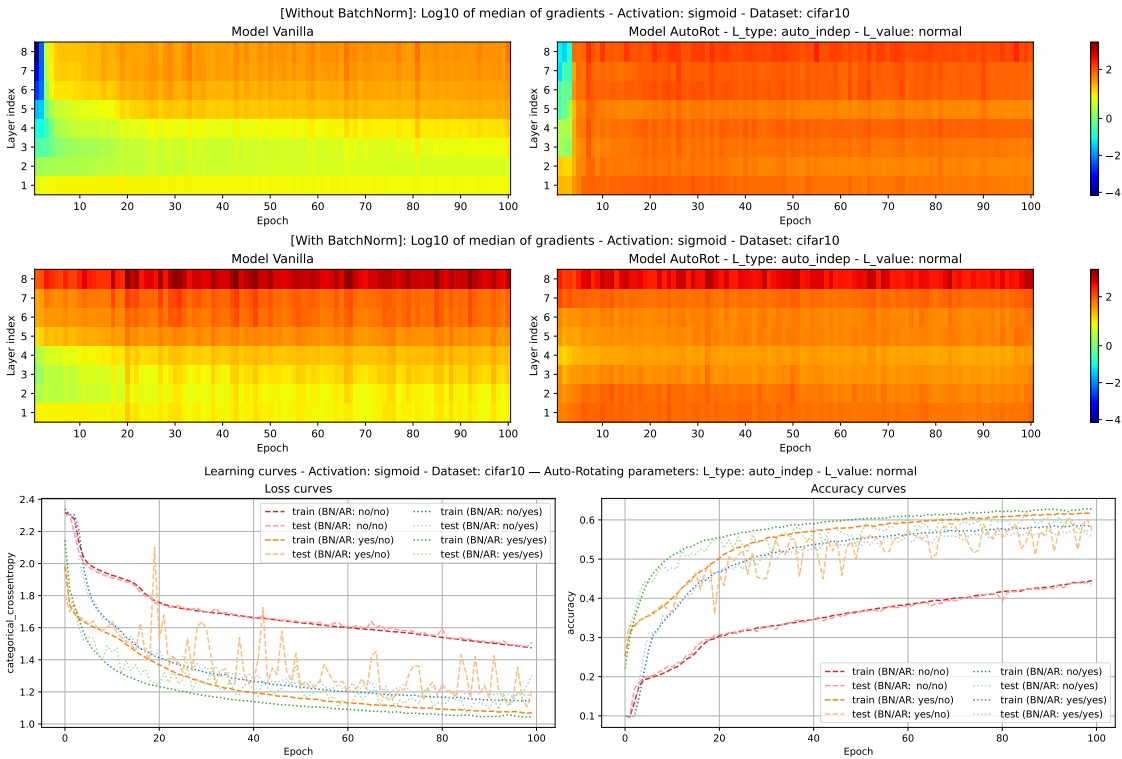

Figure 5: Gradient heatmaps and learning curves of four models with the same initial weights. Layers closer to the network input have a lower index. CIFAR-10. Arch.: $\mathcal{B}$. Activ.: sigmoid. $L$: normal.

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

# A  Appendix

## Experimentation with the Neural Architecture $\mathcal{A}$

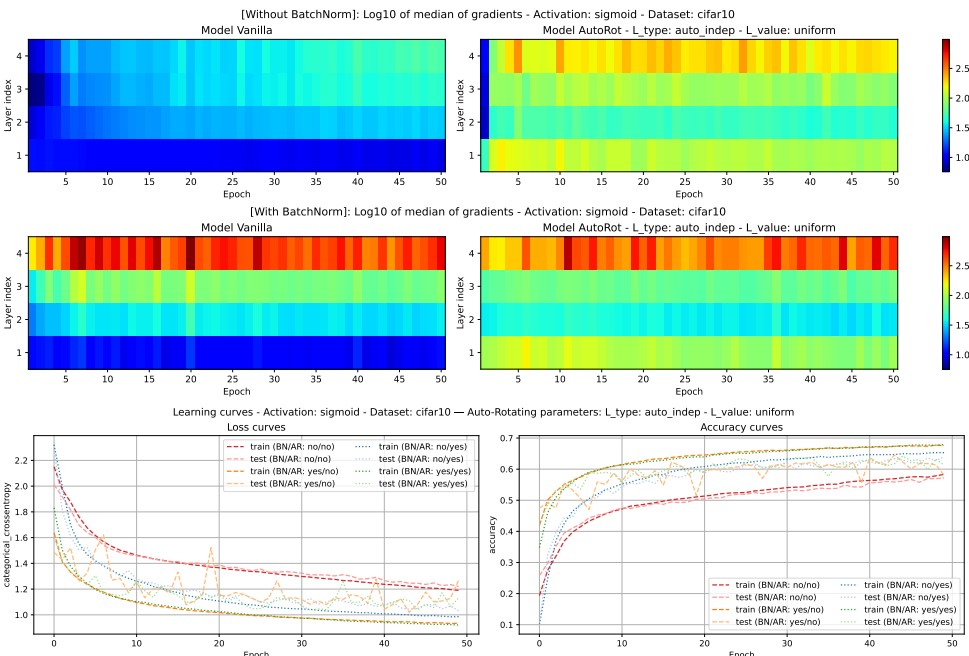

Figure 6: Gradient heatmaps and learning curves of four models with the same initial weights. Layers closer to the network input have a lower index. CIFAR-10. Arch.: $\mathcal{A}$. Activ.: sigmoid. $L$: uniform.

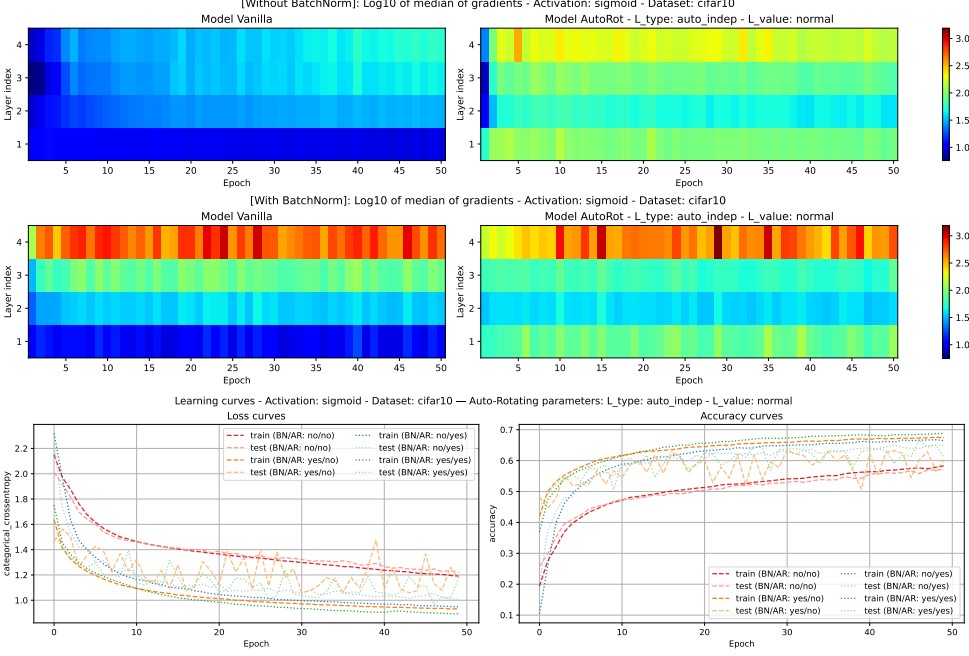

Figure 7: Gradient heatmaps and learning curves of four models with the same initial weights. Layers closer to the network input have a lower index. CIFAR-10. Arch.: $\mathcal{A}$. Activ.: sigmoid. $L$: normal.

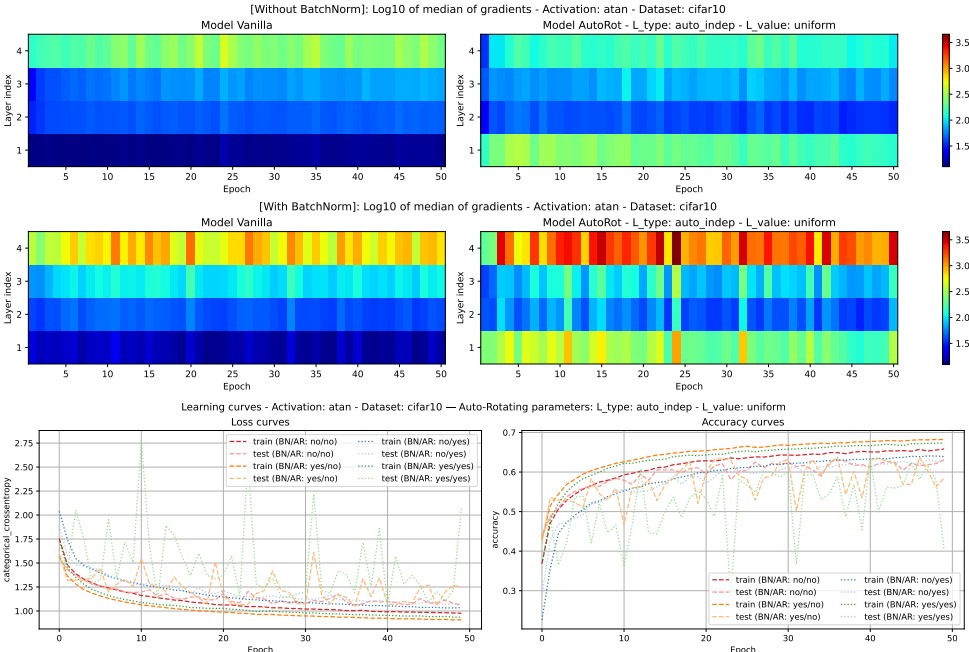

Figure 8: Gradient heatmaps and learning curves of four models with the same initial weights. Layers closer to the network input have a lower index. CIFAR-10. Arch.: $\mathcal{A}$. Activ.: atan. $L$: uniform.

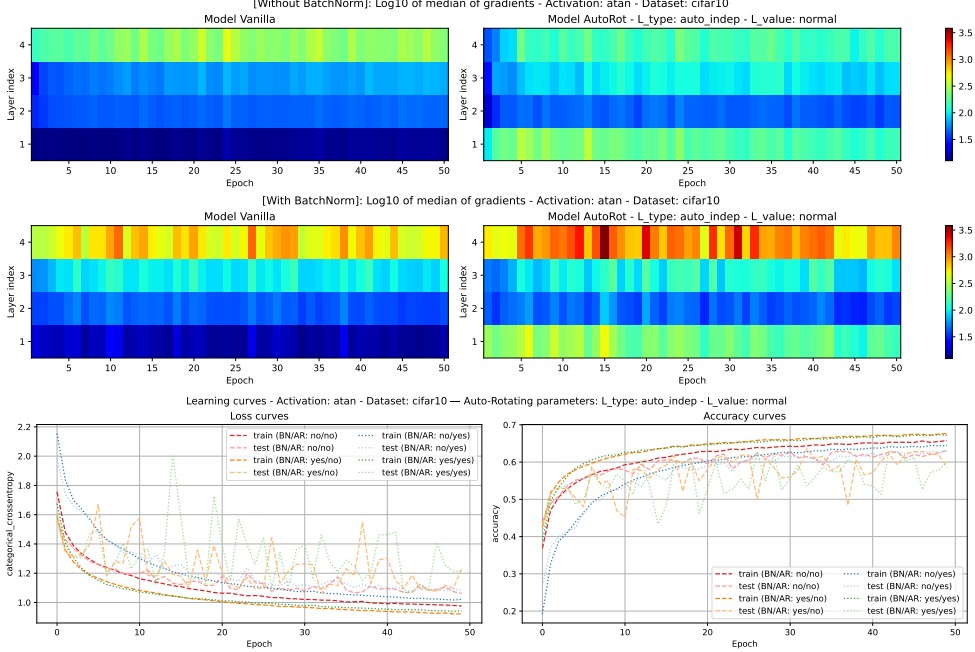

Figure 9: Gradient heatmaps and learning curves of four models with the same initial weights. Layers closer to the network input have a lower index. CIFAR-10. Arch.: $\mathcal{A}$. Activ.: atan. $L$: normal.

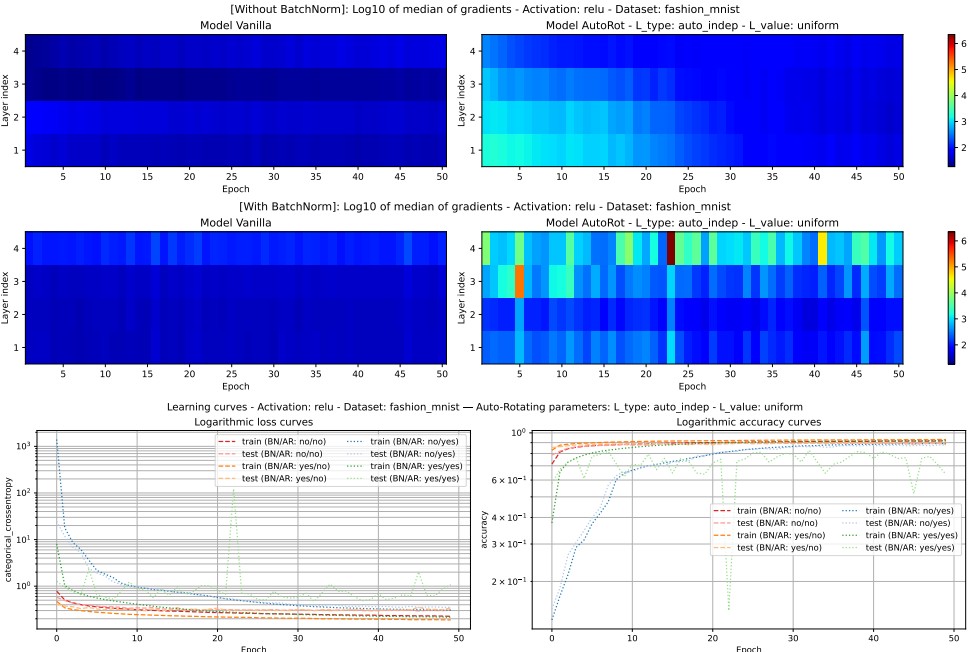

Figure 10: Gradient heatmaps and learning curves of four models with the same initial weights. Layers closer to the network input have a lower index. Fashion MNIST. Arch.: $\mathcal{A}$. Activ.: ReLU. $L$: uniform.

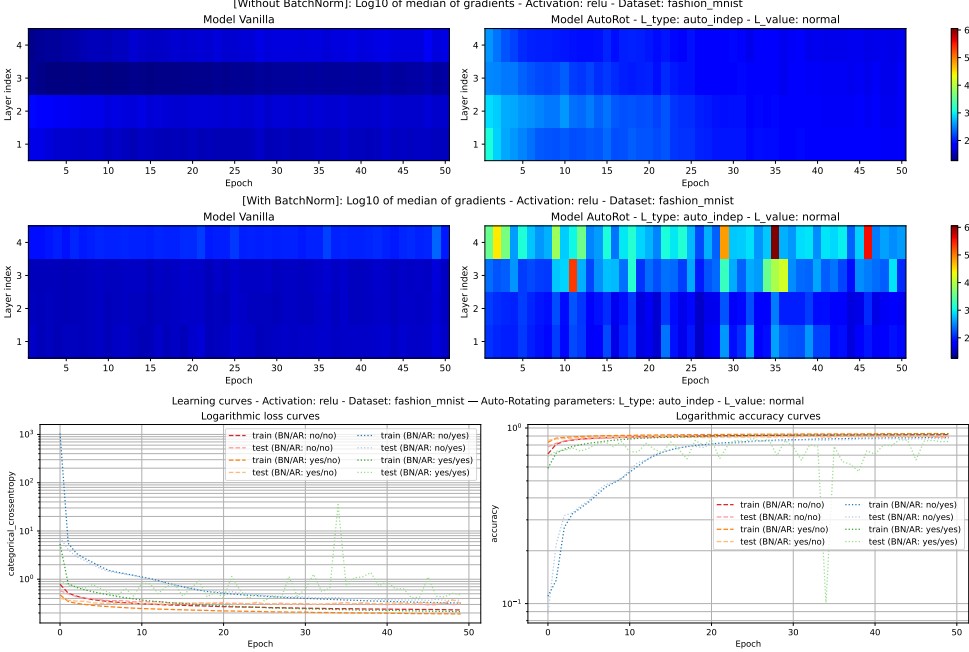

Figure 11: Gradient heatmaps and learning curves of four models with the same initial weights. Layers closer to the network input have a lower index. Fashion MNIST. Arch.: $\mathcal{A}$. Activ.: ReLU. $L$: normal.

## Experimentation with the Neural Architecture $\mathcal{B}$

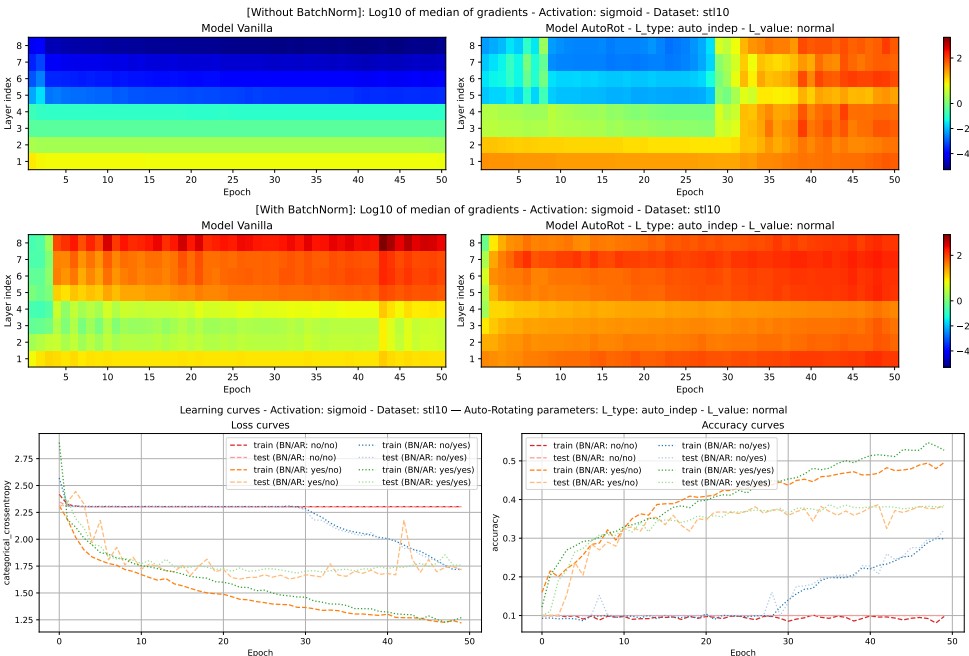

Figure 12: Gradient heatmaps and learning curves of four models with the same initial weights. Layers closer to the network input have a lower index. STL-10. Arch.: $\mathcal{B}$. Activ.: sigmoid. $L$: normal.

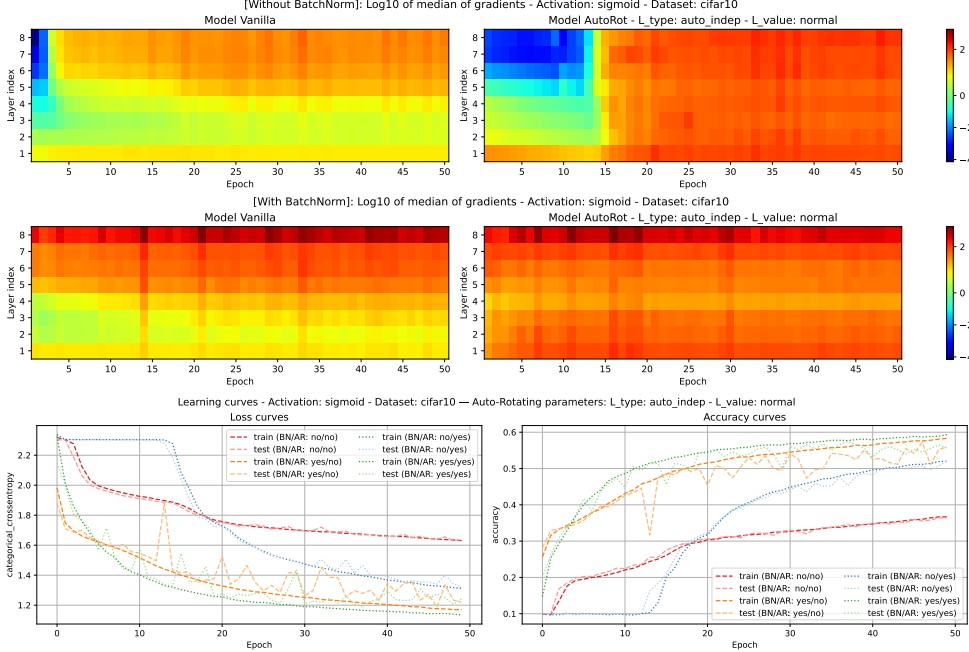

Figure 13: Gradient heatmaps and learning curves of four models with the same initial weights. Layers closer to the network input have a lower index. CIFAR-10. Arch.: $\mathcal{B}$. Activ.: sigmoid. $L$: normal.

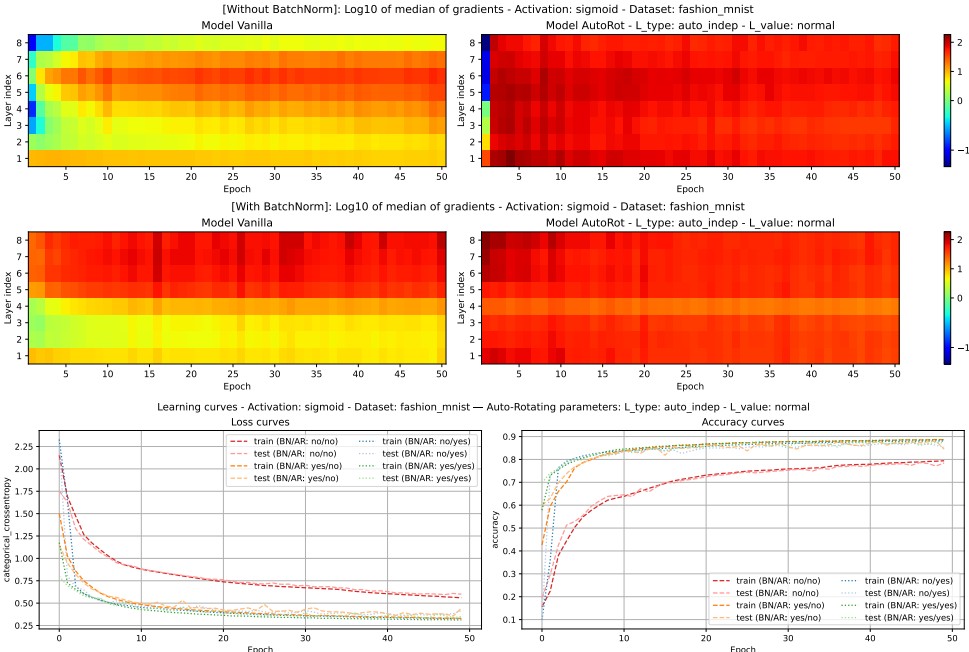

Figure 14: Gradient heatmaps and learning curves of four models with the same initial weights. Layers closer to the network input have a lower index. Fashion MNIST. Arch.: $\mathcal{B}$. Activ.: sigmoid. $L$: normal.

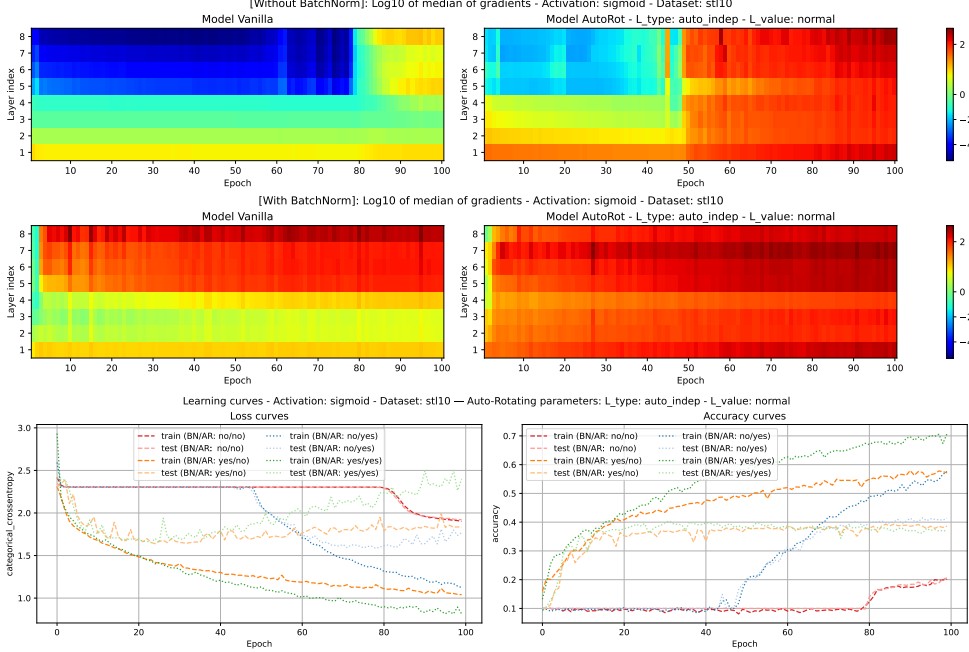

Figure 15: Gradient heatmaps and learning curves of four models with the same initial weights. Layers closer to the network input have a lower index. STL-10. Arch.: $\mathcal{B}$. Activ.: sigmoid. $L$: normal.

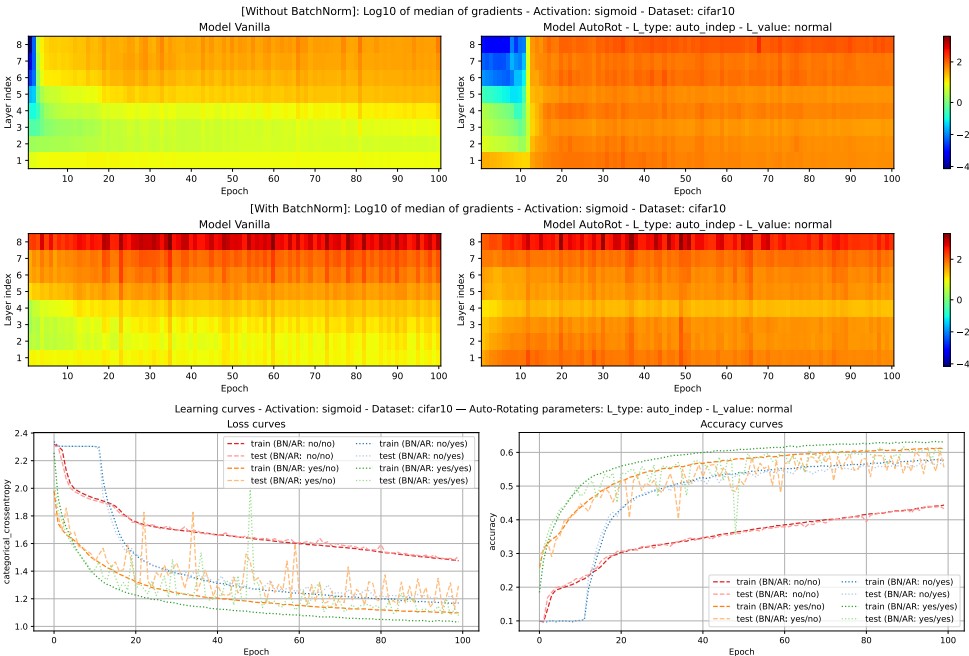

Figure 16: Gradient heatmaps and learning curves of four models with the same initial weights. Layers closer to the network input have a lower index. CIFAR-10. Arch.: $\mathcal{B}$. Activ.: sigmoid. $L$: normal.

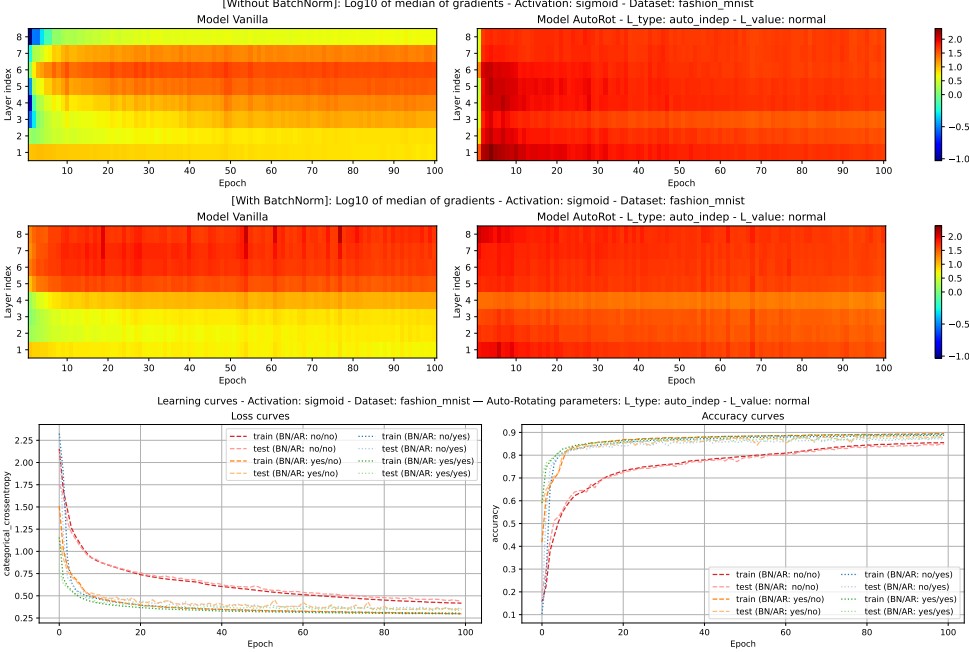

Figure 17: Gradient heatmaps and learning curves of four models with the same initial weights. Layers closer to the network input have a lower index. Fashion MNIST. Arch.: $\mathcal{B}$. Activ.: sigmoid. $L$: normal.

