# OpenReview forum: "Auto-Rotating Neural Networks: An Alternative Approach for Preventing Vanishing Gradients"
_TMLR — Withdrawn by Authors_

### Review · Reviewer_HmbH · 2023-12-12

**Summary Of Contributions:**

This work introduces a modified version of the Auto-Rotating Perceptron that normalizes the preactivation values to lie within a limited range to combat the gradient vanishing problem. In this paper, constants defining the preactivation range can be learned as neuron-specific. The approach is validated on shallow convolutional networks and several simple computer vision datasets.

**Audience:**

Yes

**Claims And Evidence:**

Yes

**Requested Changes:**

What does the $Q$ in the notation $x_Q$ denote? I think it would be better to provide the meaning of this parameter prior to giving the specific value used in the work.

What is the intuition behind the expression $x_Q = 2 x^{(min)} - x^{(max)}$? $x^{(max)} - x^{(min)} $ is the width of the activation range, but it is not clear how the $x_Q$ was deduced.

I would suggest splitting figure 3 (bottom left) into several figures since the accuracy varies drastically for datasets and the difference in performance for different values of L is hardly visible for MNIST/FashionMNIST and SVHN.

According to Figure 5, models are undertrained. I think they should be trained longer.

A general suggestion would be to conduct a study with stronger baselines and in a more challenging setup to show the benefits of the introduced method in a more practically interesting scenario.

**Strengths And Weaknesses:**

The problem of gradient vanishing and the improvement of neural network training dynamics are crucial for deep learning. Therefore, the problem addressed is practically interesting.

However, the contribution of this work seems to be incremental, since the base method was proposed in the cited work [1], and this work adds a couple of modifications to it: an option to have learnable scale constants for each channel and parametrization enforcing condition $L > 0$. Extension to convolution layers is trivial since a convolutional layer can be thought of as a linear layer applied to patches of an input image.

All experiments are performed on small-scale datasets (MNIST, Fashion MNIST, CIFAR-10, SVHN), and one cannot be sure that the conclusions would generalize to larger networks and more challenging datasets (for instance, ImageNet-1k). Modern neural networks typically involve skip connections and normalization layers, and they are known to mitigate a gradient vanishing problem to a certain extent. Would these networks benefit from the addition of the newly introduced preactivation normalization mechanism?

The network configurations considered in the work are quite small, and the accuracy achieved on the datasets is far below that reported in the literature. The classic work [2], where ResNet was introduced, achieved > 90% accuracy on CIFAR-10 without designing an elaborate training pipeline.

[1] Saromo, Daniel, Elizabeth Villota, and Edwin Villanueva. "Auto-Rotating Perceptrons." arXiv preprint arXiv:1910.02483 (2019).
[2] He, Kaiming, et al., "Deep residual learning for image recognition." Proceedings of the IEEE conference on computer vision and pattern recognition. 2016.

---

### Review · Reviewer_28QY · 2023-12-16

**Summary Of Contributions:**

This paper attempts to address the vanishing gradient problem with a pre-activation construct. The construct is evaluated using 2 models on Fashion MNIST, MNIST, CIFAR-10, SHVN, and STL-10.

**Audience:**

No

**Claims And Evidence:**

No

**Requested Changes:**

Critical: fixing all issues identified above. The paper is difficult to read and contextualize. Clarity issues significantly detract from the paper, raising concerns that few TMLR readers would be able to benefit from the paper.

**Strengths And Weaknesses:**

Strengths:
* Approach seems straightforward to implement.

Weaknesses:
* The writing is unnecessarily verbose given the complexity of the approach. The writing would be clearer by either being formal or informal. As of now, it is difficult to see the "big picture". Figure 1 is an example: it seems like a rather precise explanation. However, upon inspection, the left is showing $\hat{y}=\sigma(\mathbf{w} \cdot \mathbf{x} + \mathbf{b})$ (a common expression for most readers) with many intermediate variables with little directional flow. On the other hand, there is not much explanation for what the two half-circles mean (it looks like a decision boundary, but it seems to be a reference to 2 phases in a neuron). The right side of the figure is the paper's approach, but it's not clear what exactly I'm looking at given the ambiguities with the left with the additional changes. What is $g(\mathbf{x})=\rho f(\mathbf{x})$ supposed to be for if $g(\mathbf{x})$ only appears where it is defined? Resolving all the variable dependencies/references is a cause of reader fatigue.
* Approach is entirely heuristic but explained as if it was derived in some rigorous way. For example: "This node dynamization (i.e., desaturation) is achieved in two ways. First, by preventing the derivative of the activation function to take very small values. Recall that, to prevent the VGP, we do not want the derivative of $\sigma'(z)$ of the activation function $\sigma(z)$ to take tiny values." Isn't this argument circular (we prevent X because we do not want X)? Then, when looking above to Section 2.2, there are 3 paragraphs that are used to explain $\sigma'(z)\approx0$, when the term itself means precisely that: "Vanishing Gradient". Going into Section 3 and Figure 2, I begin to see the approach is based off of truncation of parts of the activation function. However, there is a wall of equations, which then end with "These equations result in $x_i^{(\text{min})} \leq x_i \leq x_i^{(\text{max})}$". Is the reader supposed to understand anything more than the elements in a set are in between the minimum and maximum of that set? What is the purpose of these equations? A reader wants to know what is $L$ and how neurons are modulated with it, but the discourse is getting further and further from the fundamental idea (which should have been clear by Figure 1).
* Experiments are unmotivated or even inconsistent with the message. Vanishing gradient is a problem for larger, deep networks, but the experiments use small networks. Using ReLU and Batch Normalization are claimed to have downsides, but they are directly applied with the approach.
* Experiments are not analyzed properly nor do they seem to advance the argument much. Figure 3 is what a reader would expect is the main result of the paper, but it's 5 figures that each require much explanation. They don't use the same datasets and the axes are different. For some figures, Y-axis is "relative best loss" or "best accuracy", which is a rather specific and perhaps not robust metric. It's not clear why the bottom right subfigure is with these other plots. The results aren't summarized in the abstract, nor mentioned through the paper. Overall, the experiments feel unmotivated.

---

### Review · Reviewer_CZRV · 2024-02-02

**Summary Of Contributions:**

This paper aims at addressing vanishing gradient problem(VGP) in neural networks with saturating activations by developping the Auto-Rotating Perceptron(ARP) structure and improve its design by automatically calculating the formerly hyperparameters. The authors extend the Auto-Rotating operation to convolutional layers and conduct a detailed analysis of the types and values of dynamic regions. The experiment results show the ARP operation successfully diminishes vanishing gradients and obtain better gradient properties: more uniformity across the network layers, and more stability through the training epochs.

**Audience:**

Yes

**Claims And Evidence:**

Yes

**Requested Changes:**

1. Reconsider the technical contributions of the paper.
1. More detailed experiments and a thorough analysis of the results.

**Strengths And Weaknesses:**

**Strengths:**

The motivation is clear. This paper aims to addressing vanishing gradient problem(VGP)  of saturating activations that have a good property of generating smoother network outputs than commonly used ReLU.

**Weaknesses:**

1. The technique contribution is limitied. In my perspective, this paper merely constitutes a marginal refinement of the existing ARP method by automating the computation of its two hyperparameters. Furthermore, extending ARP to convolutional layers is a straightforward mathematical operation so I do not think it can be presented as a contribution in the Introduction section.

2. The clarification of this paper is not clear. In the section 3.3, the authors claim that "we geometrically obtained that the relation $x_{Q}:=2x^{(min)}-x^{(max)}$ allows us to automatically calculate it." However, the authors did not provide an explanation regarding how this was achieved. I suggest that the authors should thoroughly analyze this point either through theoretical considerations or experimental evidence since the acquisition of $x_{Q}$ constitutes a key contribution of this paper.

3. The experiments are not sufficiently thorough, making the results unconvincing. Batch Normalization (BN) and Rectified Linear Unit (ReLU) are widely applied in deep learning, effectively mitigating the issue of vanishing gradients in various domains. To validate the effectiveness of the proposed method in this paper, the authors should compare these methods on a more extensive and deeper network architecture, as well as more complex datasets. However, the paper only conducts experiments on two simple network structures and small datasets, which does not provide sufficient evidence to prove the effectiveness and generalizability of the proposed method.

4. Explanations for the experimental results are lacking. The authors claim that using Auto-Rotation along with Batch Normalization (BN) leads to fewer vanishing gradients. However, from the figures in the appendix, it can be observed that using Auto-Rotation along with BN does not yield better test accuracy, and in some cases, it performs even worse than using BN alone. The authors should provide an explanation and clarification for this discrepancy."

---

### Note · Authors · 2024-02-18

**Comment:**

We have decided to Withdraw this article as we would need more time to address some of the issues pointed by the reviewers, in particular scaling to larger models and show generalizability of our proposed solution.

**Withdrawal Confirmation:**

I have read and agree with the venue's withdrawal policy on behalf of myself and my co-authors.